# Titration-WB: A methodology for accurate quantitative protein determination overcoming reproducibility errors

Alice Maestri[1,2], Ewa Ehrenborg[1,2], Olivera Werngren[1,2], Maria Olin[3,4], Carolina E. Hagberg[1,2], Matteo Pedrelli[3,4], Paolo Parini[3,4]*

1 Division of Cardiovascular Medicine, Department of Medicine Solna, Karolinska Institutet, Stockholm, Sweden, 2 Center for Molecular Medicine, Karolinska Institutet, Stockholm, Sweden, 3 Cardio Metabolic Unit, Department of Medicine Huddinge, and Department of Laboratory Medicine, Karolinska Institutet, Stockholm, Sweden, 4 Medicine Unit of Endocrinology, Theme Inflammation and Ageing, Karolinska University Hospital, Stockholm, Sweden

☯ These authors contributed equally to this work.
* paolo.parini@ki.se

## Abstract

Western blotting (WB) is a cornerstone technique for protein detection and quantification in molecular biology. However, its semi-quantitative nature, reliance on housekeeping protein normalization, and susceptibility to technical variability often undermine data accuracy and reproducibility. To address these limitations, we introduce titration-based Western blotting (t-WB), as innovative quantitative approach that uses serial dilutions of protein samples to generate regression curves for precise protein quantification. The method mitigates common errors, including loading inaccuracies and signal saturation, by leveraging the $R^2$ value as a quality control metric and calculating the regression line. Its slope is then used as a measure of protein concentration, expressed as signal intensity/total protein loaded. The advantage of t-WB is the removal of housekeeping protein normalization, eliminating thus the bias created by experimental conditions that may alter housekeeping protein levels. t-WB was validated across diverse setups, demonstrating its robustness in minimizing inter-experiment variability and improving accuracy by normalizing to a single internal control. By standardizing workflows, t-WB ensures reproducibility, uncovers subtle biological changes, and resolves biases inherent to classical WB protocols.

## Introduction

Western blot (WB) is a widely adopted technique for detecting and quantifying proteins in biological samples, providing essential insights into protein expression and regulation [1]. Despite its versatility, WB quantification is often limited by several challenges that reduce its accuracy and precision. One of the primary limitations is the reliance on housekeeping proteins for normalization, which introduces variability due to

**Data availability statement:** All relevant data are within the paper and its Supporting Information files.

**Funding:** In addition, we would like to clarify our financial disclosure. For both of our funding sources (P.P. 20210622;20200668;20190544 from Swedish Heart-Lung Foundation and P.P. 201602992 from Swedish Research Council), the founders had no role in study design, data collection, or preparation of the manuscript.

**Competing interests:** The authors have declared that no competing interests exist.

differences in expression levels under various conditions. Additionally, inconsistencies in protein loading, transfer efficiency, and antibody specificity can further complicate the quantification process, particularly when aiming for high accuracy [2–4]. Various approaches have been proposed to improve WB quantification, such as using total protein normalization, optimizing antibody performance, and introducing computational models to correct for potential errors [5–8]. However, most of these strategies still rely on relative measurements that are susceptible to error accumulation, and none provide a robust, universally applicable method for protein quantification.

To address these limitations, we introduce the t-WB method, which combines part of the principles of classical titration assays used in chemistry, with the WB technique. Titration assays involve systematically varying the concentration of a substance of interest and measuring the subsequent change in signal output intensity to determine the concentration of the analyte. In t-WB, we expand this concept by generating a function that relates the signal intensity of the protein of interest to the amount of total protein loaded onto the gel at multiple concentrations. The first-order derivative of this function is then used to quantitatively estimate the target protein concentration [9]. While not offering absolute quantification of protein concentration, which would require standards such as recombinant proteins of known concentrations, t-WB still provides significant improvements in accuracy and precision over classical WB techniques. It does so by eliminating many of the errors introduced by normalization to housekeeping proteins or total protein mass (such as signal saturation of the housekeeping protein, variable expression due to known or unknown phenomena, narrow dynamic range and detection sensitivity and membrane compatibility, to name a few), offering a more reliable and reproducible approach for protein quantification expressed as signal intensity (A.U.) per total protein mass loaded (μg). In this work, we outline the development, standardization, and validation of t-WB, showcasing its applicability especially for challenging experimental setups, for example for evaluating proteins with very high or very low molecular weights, with various post-translational modifications, or with high reactivity to experimental conditions such as lipid treatment. As t-WB has already been employed both in clinical and preclinical studies [10–16], this work aims at framing it as a standardized method and expanding its uses and applications.

## Results

The t-WB pipeline is outlined in Fig 1a. Firstly, each sample lysate is prepared at three different serial dilutions, allowing the loading of three increasing total protein masses of the same sample onto the WB gel. The dilution range must be carefully optimized to ensure that immunodetection is possible across all dilutions while achieving an optimal signal-to-background ratio and maintaining linearity without reaching saturation, as also previously suggested [17,18]. The use of at least three dilutions for each specimen represents a fundamental difference compared to classical western blotting, where each specimen is typically tested by loading only a single protein mass. The processes of gel electrophoresis, transfer, and immunodetection follow standard operating WB procedures. From the serial dilutions of each sample,

**Fig 1. t-WB: pipeline and detection of immunoblotting errors.** a) Schematic showing the t-WB pipeline. b) Schematic showing an ideal t-WB: WB signal (upper insert) shows increasingly thick bands for increasing total protein mass loaded, for which the plotted signal results in a linear curve with a R2 of 1. The line of best fit is described and its first derivative is 6 (middle), thus the quantification of the protein of interest is 6 signal intensity units

(AU)/total protein mass (µg), shown in the bar plot (right). c) Representative immunoblot of an ideal t-WB, detecting LDLR in HepG2 cells. Total protein loaded were 20, 40 and 60 µg. d) t-WB analysis of (c) with t-WB plot (left), line of best fit and its derivative (middle) and LDLR quantification shown in the bar plot (right). e) Schematic showing a saturated t-WB: WB signal (upper insert) shows thick bands for increasing total protein mass loaded, for which the plotted signal results in a linear curve with a R2 less than 0.9. The line of best fit is described and its first derivative is 4.6 (middle), thus the quantification of the protein of interest is 4.6 signal intensity units (AU)/total protein mass (µg), shown in the bar plot (right). f) Representative immunoblot of a saturated t-WB, detecting PLIN3 in THP-1 cells. Total protein loaded were 5, 10 and 20 µg. g) t-WB analysis of (f) with t-WB plot (left), line of best fit and its derivative (middle) and LDLR quantification shown in the bar plot (right). h) Schematic showing errors in total protein content quantification or changes in antibody affinity/avidity in 3 repeated t-WBs: WB signals (upper insert) show thick bands for increasing total protein mass loaded, for which the plotted signal results in 3 linear curves. The lines of best fit are described, and their first derivatives are 6 (middle), thus the quantification of the protein of interest among these three blots is maintained at 6 signal intensity units (AU)/total protein mass (µg), shown in the bar plot (right). i) Representative immunoblots of two t-WBs with different antibody affinity/avidity, detecting HSC70 in THP-1 cells. Total protein loaded were 5, 10 and 20 µg. h) t-WB analysis of (i) with t-WB plots (left), lines of best fit and their derivatives (middle) and HSC70 quantification shown in the bar plot (right).\\t-WB: pipeline and detection of immunoblotting errors.

three separate intensity signals are obtained for that sample. To obtain the results, these intensity signals are plotted against their corresponding total protein mass of loaded specimen. Since we operate under non-saturating conditions, the resulting XY graph is evaluated using the *least squares method*, and the line of best fit for the regression is calculated (Fig 1a).

The schematic in Fig 1b shows an ideal t-WB: the three sample dilutions result in three bands, whose intensities are plotted against the correspondent total protein mass loaded for that sample. The resulting curve has an $R^2$ of 1, which affirms there was no loading error of the sample nor a saturation of the signal (Fig 1b; S1 Table a). To obtain a quantitative measurement of the protein of interest, the first derivative of the function $f(x) = ax+b$ describing the regression line is calculated. The *a,* also commonly referred as slope, estimates the concentration of the target protein, expressed as arbitrary units (A.U.) of signal intensity *per* mass (µg) unit of total protein loaded, and thereby gives a quantitative measurement of the concentration of the protein of interest. In our schematic example of an ideal t-WB, *a* corresponds to 6 signal intensity/µg of total protein loaded. A real-world example of a t-WB, shown in Fig 1c, demonstrates quantification of low-density lipoprotein receptor (LDLR) expression in a HepG2 cell lysate. When the detected signal intensities are plotted against the corresponding total protein mass loaded (20, 40 and 60 µg), the first derivative of the regression line is calculated to be 2.14 (Fig 1d; S1 Fig; S1 Table b). Consequently, the LDLR concentration in the sample is 2.14 A.U. of signal intensity/µg of total protein loaded. Notably, by using t-WB, we could exclude major loading errors as well as signal saturation as the $R^2$ of the regression line is 0.96.

In classical WB workflows, errors such as inaccurate sample dilution, improper gel loading, and signal saturation are often challenging to both detect and resolve. Repeating the analysis by running a new identical WB often fails to address the source of the errors, as the same methodological limitations would persist. In contrast, the t-WB approach leverages the $R^2$ value as a key metric to assess sample accuracy and detect errors. Decreases in $R^2$ below 0.9 indicate operator errors, such as inaccurate sample loading, and/or signal saturation across the diluted samples (Fig 1e; S1 Table c). As a practical example, an attempt to measure the expression of PLIN3 is shown (Fig 1f,g; S2 Fig. a), where both the regression line and the resulting low $R^2$ value reveal a lack of linearity, pointing to improper sample loading and showing the utility of t-WB (Fig 1g; S1 Table d).

The use of linear regression also allows the study of the intercept *b* (Fig 1a), which here reflects the contribution of background to the intensity measured in each point, and thereby informs on the signal-to-background ratio. The intercept *b* also depends on the affinity/avidity of a primary antibody for the protein of interest, which is influenced by the co-localizing proteins present at the target site or by effects of the varying WB procedures on the epitopes. Changes in antibody affinity/avidity are reflected in shifts of the regression line vertically along the y-axis (Fig 1h; S1 Table e). Additionally, discrepancies in *b*-values between replicates (e.g., two control samples analyzed in separate t-WBs) may indicate an error in quantifying the total protein concentration of the sample stock solution prior to titration. Such errors cause the

regression lines to shift horizontally along the x-axis: leftward if the protein concentration is underestimated and rightward if overestimated. However, since the operator is not aware of this error and does not plot the real amount of total protein loaded, the shift will be visualized on the y-axis instead (Fig 1h). In t-WB, these sources of bias do not affect the final quantification of the protein of interest because the slope remains unaffected, ensuring high accuracy in quantification (Fig 1h, right). A real-life example where the affinity/affinity of the antibody has changed is shown in Fig 1i using t-WB to analyze HSC70 (S2 Fig. b). The same cell lysate sample was loaded in three total protein masses twice on the same WB gel and separated by electrophoresis. The membrane was then divided in two (termed P1 and P2, respectively) prior to the antibody incubation-step. For P1, the membrane was directly incubated with the HSC70 antibody, while P2 was first incubated with a PLIN2 antibody (not shown) and then re-probed for HSC70. The reduced HSC70 antibody signal intensity in P2 highlights the variation in antibody affinity/avidity that frequently arises in Western blotting when membranes are probed and re-probed in different sequences. In a classical WB setup, this could lead to inaccurate quantification of the protein of interest if the changes in affinity/avidity are not proportional between the antibody used for detection of the protein of interest, and antibody used to detect the housekeeping protein. As demonstrated in Fig 1j and in the S1 Table f, in t-WB these experimental limitations have a relatively low impact on the final quantitative measurement of HSC70 leading to largely the same result.

To evaluate the precision of the t-WB method, we compared signals from six independent t-WB blots measuring LDLR expression in HepG2 cell lysates, WB1 shown in Fig 1c and WB2–6 in Fig 2a (Fig 1c, Fig 2a-2e; S1 Fig.). The first derivatives of the regression lines for each blot were analyzed, and the coefficient of variation (CV) for the slopes was calculated, resulting in a CV of 36% (Fig 2c; S2 Table a). This variability among blots is common and arises due to several factors, including differences in background signal, loading discrepancies, and technical variations between independent WB runs, which together lead to differences in the slopes of the regression lines. To reduce this variability, we chose WB1 (shown in Fig 1c) as reference blot, and normalized the all signals on the remaining blots (shown in Fig 2a) to each of the signals from the reference blot (corresponding to 20 µg, 40 µg, and 60 µg loaded protein mass, respectively). This normalization led to a reduction in the CV: from 36% to 18% when using the signal at 20 µg total protein loaded (S3 Fig. a,b; S2 Table b), to 12% when using the signal at 40 µg protein loaded (S3 Fig. c,d; S2 Table c), and to 4.2% when using the signal at 60 µg protein loaded (Fig 2d, 2e; S2 Table d). To further confirm the reproducibility of this technique, we repeated the analysis using a different primary antibody targeting the protein LAMP2A instead (Fig 2h; S4 Fig.). We again observed a similar reduction in CV, from 52% (Fig 2h; S3 Table a) to 8.5% (Fig 2i,2j; S3 Table b-S3d) when normalizing all t-WB signals to the LAMP2A signal at 60 µg total protein loaded from WB1 (Fig 2f). t-WB also allows us to assess protein quantity in WBs with lower quality, for example in WB2 where there is a visible bubble. While in regular WBs with a single point we wouldn't have been able to utilize this data, the use of three total protein dilutions of the same sample and the t-WB analysis with its one-point normalization highlighted the relative insignificant interference that this bubble had on the quantification of LDLR and LAMP2A (Fig 2e, 2j). To confirm the reproducibility of the assay we repeated the t-WB analysis using a different cell lysate, running four independent WBs, each detecting both the LDLR (S3 Fig. i-m; S5 Fig.) and LAMP2A (S3 Fig. n-r; S6 Fig.) proteins. Similarly to the previous examples, normalization of the signals from the four blots reduced the CV from 8.3% to 4.9% for LDLR (S4 Table a,b), and from 46% to 6.1% for LAMP2A (S4 Table c,d) when using the signal at 60 µg total protein loaded from WB1 (S3 Fig. i,n, respectively). These results demonstrate that normalizing the signal intensity to a reference sample with the highest signal-to-background ratio, in our case the sample with 60 µg of total loaded protein, allows for more precise and reliable comparisons across multiple blots and experiments.

The fundamental advantage of the t-WB method is that it allows the quantitative measurement of target proteins without the need for normalization to a housekeeping protein. Traditional normalization to proteins like β-actin, GAPDH, or tubulin is prone to inaccuracies, as these housekeeping proteins can be affected by experimental conditions such as cell growth, gene knockdown or treatment, resulting in significant over- or underestimation of the protein of interest. To demonstrate

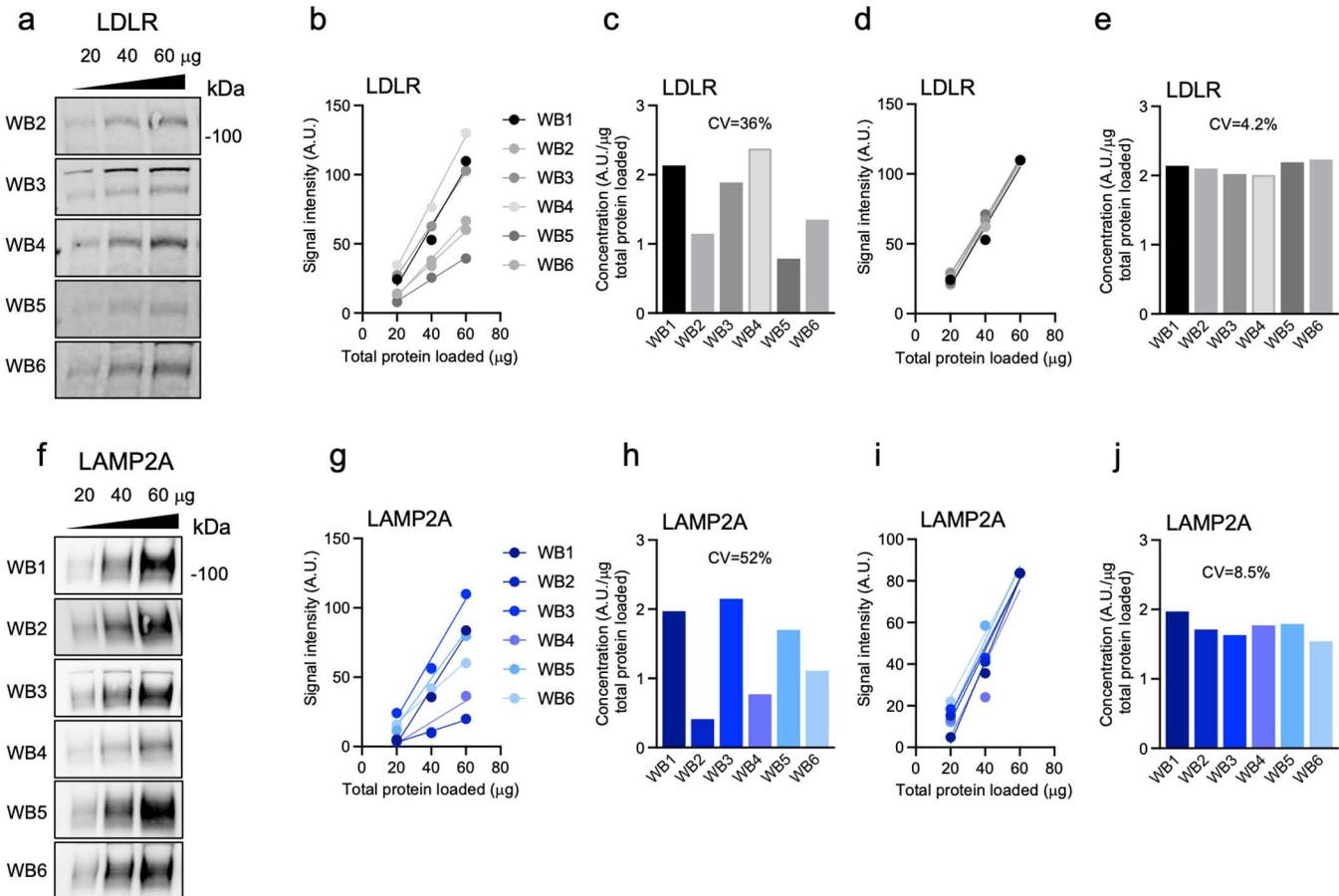

**Fig. 2. Normalization to a single dilution point reduces the coefficient of variations given by blot-to-blot variance.** a) Immunoblots of a single lysate of HepG2 cells, loaded at 20, 40, 60 µg of total protein onto 6 different WB membranes and probed for the LDLR. b) t-WB plots for LDLR for each WB1-6. c) LDLR concentrations from each WB expressed as signal intensity units (AU)/total protein mass (µg) and the overall CV. d) t-WB plots for LDLR after normalization of all data points using the WB1-60 µg data point. e) LDLR concentrations from each WB from (d) expressed as signal intensity units (AU)/total protein mass (µg) and the overall CV. f) Immunoblots the same 6 WB membranes shown in (a) with LAMP2A antibody. g) t-WB plots for LAMP2A for each WB1-6. h) LAMP2A concentrations from each WB expressed as signal intensity units (AU)/total protein mass (µg) and the overall CV. i) t-WB plots for LAMP2A after normalization of all data points using the WB1-60 µg data point. j) LAMP2A concentrations from each WB from (i) expressed as signal intensity units (AU)/total protein mass (µg) and the overall CV.

this, and how t-WB can be used to mitigate it, we evaluated the expression of the lipid droplet-associated protein perilipin 2 (PLIN2) in THP-1 cells under various lipid loading conditions (Fig 3a; S7 Fig. a; S8 Fig).

Using t-WB, we plotted the detected signal intensities for PLIN2 from three different dilutions for each condition against the respective protein loads (Fig 3b,3c). The slopes of the regression lines were calculated and used to quantify the effects of lipid treatments (Fig 3c; S5 Table a). The results showed that treatment with either 25 or 50 µg/mL oxidized low-density lipoprotein (oxLDL) increased PLIN2 expression similarly as compared to control, while oleic acid (OA) significantly boosted PLIN2 levels only slightly higher (Fig 3c; S5 Table a). To demonstrate the limitation of using classical WB and housekeeping protein normalization, we first quantified the commonly used housekeeping protein β-actin by t-WB. The analysis showed that β-actin expression was reduced by 20% by the OA treatment (Fig 3d; 3e; S8 Fig.; S5 Table b). This effect was not due to loading errors or mistakes, proven by the very high $R^2$ (0.98) in the t-WB regression for PLIN2 (S5 Table a). When data for 2.5 µg total loaded protein mass was used and the PLIN2 signal was normalized for β-actin,

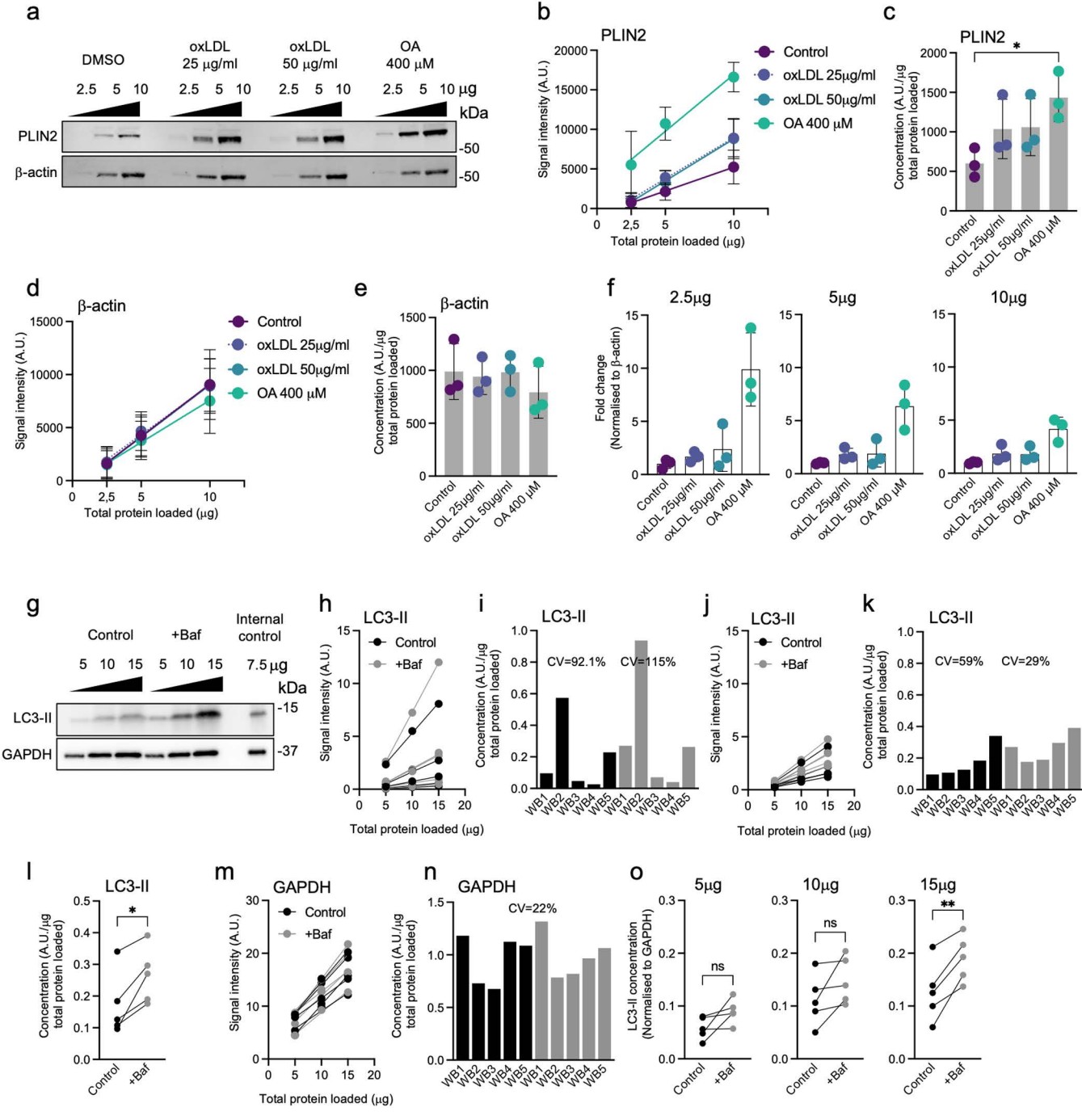

**Fig. 3. Enhanced accuracy of *t*-WB protein expression quantification as compared to classical WB analysis.** a) Representative immunoblots of PLIN2 and β-actin of THP-1 cells treated with vehicle (DMSO), 25 μg/ml oxLDL, 50 μg/ml oxLDL or 400 μM OA. Each cell lysate sample was loaded at 2.5, 5, 10 μg of total protein per well. **b**) t-WB plots for PLIN2 in each condition. c) PLIN2 concentrations expressed as signal intensity units (AU)/total protein mass (μg). Data presented as ($n_{exp}$ = 3) indicated by the data points. Bars represent mean ± s.d. p values refer to ns (not shown), *p < 0.05; (one-way ANOVA with Dunnett's multiple comparisons test). **d**) t-WB plots for β-actin in each condition. e) β-actin concentrations expressed as signal intensity units (AU)/total protein mass (μg). f) Relative PLIN2 quantification expressed as fold change of PLIN2 normalized to its respective β-actin signal. Data is shown separately for each total protein load. g) Representative immunoblots of LC3-II and GAPDH of THP-1 cells treated with vehicle (DMSO) or Bafilomycin (+Baf). Each cell lysate sample was loaded at 5, 10, 15 μg of total protein per well. For each WB, a common internal control of THP-1 cells was loaded at 7.5 μg. **h**) t-WB plots for LC3-II in each condition. i) LC3-II concentrations expressed as signal intensity units (AU)/total protein mass (μg)

and the CVs per treatment. j) t-WB plots for LC3-II after normalization of all data points using the common internal control (reference WB1). k) LC3-II concentrations from each WB from (j) expressed as signal intensity units (AU)/total protein mass (μg) and the CVs per treatment. l) LC3-II concentrations grouped as control vs treatment. Data presented as ($n_{exp}$ = 5) indicated by the data points. Paired values are shown. $p$ values refer to *$p < 0.05$; (unpaired two-tailed t-test). m) t-WB plots for GAPDH in each condition. n) GAPDH concentrations expressed as signal intensity units (AU)/total protein mass (μg) and the overall CV. o) LC3-II concentration expressed LC3-II/GAPDH. Data is shown separately for each total protein load. Data presented as ($n_{exp}$ = 5) indicated by the data points. Paired values are shown. p values refer to ns p > 0.05, **$p < 0.01$; (unpaired two-tailed t-test).

the effect of OA on PLIN2 became grossly overestimated (9.9-fold increase vs. 2.4-fold with t-WB) (Fig 3f, first panel; S5 Table c). Using the data for higher levels of loaded protein (5 μg and 10 μg) led to a more accurate quantification (6.4-fold and 4-fold, respectively; S5 Table d,e), but even then, the measurements did not accurately capture the OA-induced effect on PLIN2 expression. In conclusion, t-WB allows for accurate quantification of target proteins such as PLIN2 without relying on potentially variating housekeeping proteins, eliminating biases introduced by conventional normalization methods and providing more reliable results.

Lastly, we aimed at demonstrating the utility of t-WB for analyzing samples from independent experiments analyzed on different WBs. LC3-II protein levels were quantified in THP-1 cells treated with bafilomycin (Baf), an autophagy inhibitor that induces LC3-II accumulation. Five independent experiments were conducted, with cell lysates loaded onto separate gels along with an internal control that was loaded onto every gel (at a concentration of 7.5 μg total protein) to enable cross-experiment normalization (Fig 3g; S7 Fig. b; S9 Fig.). Without normalization, LC3-II slopes exhibited high variability, with CV of 92.1% and 115% for control and Baf-treated samples, respectively (Fig 3h, 3i; S6 Table a). Normalization using the internal control (reference blot WB1) reduced the CV to 59% and 29%, respectively, allowing reliable comparisons and detection of the expected increased LC3-II accumulation in Baf-treated cells as compared to controls (Fig 3j-3l; S6 Table b). To compare t-WB results to those using classical WB analysis, we again first assessed the housekeeping protein GAPDH by t-WB. Upon normalization using the shared internal control, GAPDH slopes showed a CV of 22% and no treatment-specific effects (Fig 3m, 3n; S7 Table a,b). LC3-II was then quantified via standard WB analysis, normalizing its signal to that of GAPDH separately of each level of total protein loaded to the gel (Fig 3o). This showed that while higher levels of loaded protein onto the WB gel (15 μg) produced the expected results, lower levels of total protein (5 μg, 10 μg) only resulted in a non-significant trend of LC3-II protein expression after Baf treatment (Fig 3o; S7 Table c-e). Taken together, we here demonstrate the utility of t-WB for analyzing samples from independent experiments run on different WBs. With the aid of a shared internal control, we were able to normalize the slopes similarly as in Fig 2 and thus to generate consistent results independently of total protein loaded and more reliably than when normalization to the expression of a housekeeping protein.

## Discussion

In this study, we present t-WB as a novel and reliable approach for improving the accuracy and precision of protein quantification in Western blotting. We demonstrate that t-WB eliminates the reliance on housekeeping proteins and offers a straightforward means of detecting pipetting errors, as well as enabling robust normalization across different blots and experiments. By addressing key sources of error inherent in classical Western blot analysis, t-WB provides a powerful solution for researchers seeking more reliable and quantitative protein detection.

The core principle of t-WB draws from the titration assay commonly used in chemical analyses to establish a relationship between the concentration of a substance and its measurable effect. In the context of t-WB, this relationship is between the concentration of the target protein and the corresponding signal intensity. The t-WB plot enables the operator to instantly visualize saturation, pipetting errors and affinity/avidity changes. The latter can be assessed by analyzing the intercept (or *b*-value) of the line of best fit. For an ideal blot (Fig 1b), all the *b*-values obtained, either from different samples run within the same t-WB, or from the same sample run as different t-WB replicates, would be the same and

approaching zero. This is assuming that: 1) the background does not influence the signal detection; 2) the affinity/avidity of the antibody towards the protein of interest in the different specimens does not change; and 3) the loading of the specimen and its total protein mass are accurate. Pillai-Kastoori and colleagues described that the linear relationship between sample concentration and band intensity is lost at low and high total protein mass loaded, since the tail and shoulder end of the intensity readout curve are strongly affected by noise and signal saturation, respectively [18]. In line with this, Pitre and colleagues also showed that the physical relation between the incident and emergent energy of light traversing a solid substance (i.e., a WB band on a film) is logarithmic and not linear, due to the Beer-Lambert law [17]. This is seen in practice in many of the examples presented in this study, where the *b*-value is shown to be altered by re-probing (Fig 1j) or different blotting conditions (Fig 2).

Theoretically, the affinity and avidity of an antibody should remain constant as they both are intrinsic antibody properties. Affinity refers to the strength of interaction between the antibody's antigen binding site and its epitope, while avidity represents the overall strength of the antibody-antigen interaction. However, in practice, the physical presence of co-localizing proteins on blot membranes can affect affinity and, consequently, also antibody avidity. This occurs through mechanisms such as steric hindrance, epitope masking, conformational changes, crowding effects, and non-specific interactions. The phenomenon is crucial to consider in protein analysis, but often overlooked since it is very hard to detect by loading and analyzing a single protein sample, as the high variance across blots will mask and confound it. Moreover, if the signal intensity of the protein of interest is affected, but not that of the housekeeping protein due to the use of two different antibodies, this will result in gross inconsistencies between experiments and under/overestimation that are hard to recognize. In this report we show t-WB may offer both the ability to recognize such inconsistencies, and the possibility to quantify the protein of interest without relying on a housekeeping protein.

While classical WB typically relies on normalization to housekeeping proteins or total protein mass of the sample, t-WB evades the erroneous assumption that the expression of housekeeping proteins is not affected by experimental conditions. Instead, t-WB uses the mathematical relationship between signal intensity and protein concentration across multiple wells loaded with the same sample to estimate the concentration of the target protein with greater accuracy. Several researchers have previously suggested the use of either calibration curves, serial dilutions and sample mixtures as standard curve in WB experiments to improve the outcome of the method [5,7,17,19,20]. However, previously proposed approaches only yield a relative, semi-quantitative quantification of protein content, while still relying on housekeeping proteins and/or normalization to total protein signal, which may lead to data misinterpretation, as demonstrated by our work and by others [21–23]. Thus, the t-WB framework represents a significant new advancement of WB quantification in terms of applicability, robustness, and precision by completely eliminating the use of housekeeping proteins for normalization.

Another key advantage of t-WB is its ability to improve the comparison of experimental groups run on different blots by minimizing the bias that arises from differences in sample loading and WB preparation. With t-WB, we were able to normalize protein signals across different experiments, even when performed on separate gels, making inter experiment comparisons more reliable. This normalization between experiments, in combination with the use of internal controls, further enhanced the reproducibility of results, even under different experimental conditions. Even when variance of the housekeeping protein is limited, as shown in Fig 3m, the t-WB result was still more reliable than classical WB normalization, which showed a conclusive result only when high levels of protein lysate were loaded (Fig 3o). It is important to note, however, that while t-WB facilitates the quantification of relative protein levels between samples, it does not provide absolute protein concentrations. This limitation arises from its reliance on signal intensity relative to the total protein mass loaded, as well as the absence of specific standards for the protein of interest. Despite this caveat, shared with classical WB analyses, t-WB represents a notable advancement in protein quantification, and is likely to contribute to improved reproducibility in published scientific results.

To test the consistency of t-WB and its applicability to different laboratory setups and methodologies, we have performed the experiments shown in this report using different WB techniques, from precast gels to homemade gels, wet

transfer to semi-dry, nitrocellulose membranes and PVDF, chemiluminescence and fluorescent signal detection, and by detecting the signals using either LI-COR or ChemiDoc machines. For all different setups and settings, application of t-WB produced more accurate and less variable results than classical WB. A clear trade-off for t-WB is the requirement for a higher amount of biological material to perform the titration, and the need for more wells and blots per experiment. However, this investment is offset by the significant advancements the method offers.

We conclude that t-WB provides a more reliable and reproducible method for quantifying protein levels across multiple experiments. By minimizing variability and improving cross-experiment normalization, t-WB enhances the accuracy of protein quantification compared to traditional normalization methods, such as using housekeeping proteins, total protein normalization or their combination. These findings highlight the utility of t-WB for more precise and consistent quantification, especially for complex experimental setups where the expression of the housekeeping protein is likely changed.

## Conclusion

The titration-Western Blot (t-WB) method represents a significant advancement in protein quantification, directly addressing and overcoming the persistent limitations of traditional Western blotting. By seamlessly integrating combining titration analysis with regression-based normalization, t-WB establishes a gold standard for accurate, reproducible and truly quantitative protein measurements. Unlike conventional approaches, t-WB eliminates the dependence on housekeeping proteins and neutralizes biases introduced by experimental variability, making it an exceptionally reliable and standardized solution for protein analysis. The use of serial dilutions and regression analysis enables precise quantification, while built-in quality control metrics such as $R^2$ values and intercepts (b) help detect and correct experimental inconsistencies. These features provide a level of accuracy and control that distinguishes t-WB from conventional methods.

Our validation across diverse experimental conditions has shown that t-WB not only reduces inter-experiment variability, but also significantly enhances cross-blot comparisons and uncovers subtle biological changes that traditional normalization methods routinely miss. Remarkably, t-WB maintains its high performance even with low-quality Western blot runs, such as those compromised by bubbles or protein smears, underscoring its robustness and adaptability in real-world laboratory settings.

By decisively overcoming the shortcomings of classical Western blot protocols, t-WB redefines the standards for quantitative protein analysis. It equips researchers with a powerful, trustworthy tool that guarantees superior accuracy and reproducibility, fundamentally advancing the reliability and impact of molecular biology research. As scientific methodologies continue to evolve, t-WB stands as a foundational innovation, bridging the gap between qualitative detection and precise quantitative analysis. Its widespread adoption will not only elevate the rigor and reproducibility of scientific studies but also drive deeper, more reliable insights into the intricate dynamics of protein expression and regulation. The t-WB method is poised to become an indispensable asset in the pursuit of scientific excellence.

## Methods

### Cell lines and treatments

THP-1 were purchased from ATCC (#TIB-202) and cultured in RPMI 1640 medium with Glutamax supplement (#61870036, Thermo Fisher Scientific) and 10% heat inactivated foetal bovine serum (FBS; # A5669801, Thermo Fisher Scientific). Cells were seeded in 6 well-plates, at 37°C in a 5% $CO_2$ atmosphere and differentiated to macrophage-like cells with 50 ng/ml phorbol myristate acetate (PMA; #P1585-1MG, Merck Life Science AB) for 24h, then rested for 48h prior to treatments. Cells were treated for 24h with DMSO (control), oxLDL (25 or 50 μg/ml, isolated following [24]), OA (400 μM), bafilomycin A1 (100 nM; #54645 Cell Signaling Technologies). Cells were washed twice with cold PBS, then harvested using a scraper in lysis buffer (50 mM HEPES pH 8,0, 150 mM NaCl, 1% NP-40). Protein isolation in lysis buffer was performed for 15 min in ice, then centrifuged at 16,000 x g for 15 min at 4°C. The collected supernatant (total protein) was stored at −20°C. The total protein content was quantified by Pierce BCA Protein assay kit (#23225, Thermo Fisher Scientific).

HepG2 were purchased from LGC Standards/ATCC (#ATCC-HB-8065). We have previously developed a protocol that improves the phenotype of HepG2 cells and make them to become more hepatocyte-like in terms of hepatic lipid metabolism (hs-HepG2; [25]. According to this protocol HepG2 cells were cultured in 6 well-plates, at 37 °C in a 5% $CO_2$ atmosphere in DMEM (1 g/L glucose; # 10567014, Thermo Fisher Scientific) supplemented with 100 units/mL penicillin, 100 µg/mL streptomycin, and 2% human serum for 10 days. Then, hs-HepG2 were incubated for 18h in Reduced-Serum Medium w/o phenol red (Opti-MEM; # 11058021, Thermo Fisher Scientific). Atorvastatin (5 mM; #3776, Tocris) treatment was administered for 16h prior to harvest. Cells were washed in cold PBS, then harvested in Pierce RIPA buffer (#89900 Thermo Fisher Scientific). Protein isolation with RIPA buffer was performed for 15 min in ice, then centrifuged at 16,000 x g for 15 min at 4°C. The collected supernatant (total protein) was stored at −80°C. The total protein content was quantified by Lowry method (DC™ protein assay, # 5000111, Bio-Rad Laboratories).

**Immunoblotting of PLIN3, HSC70, PLIN2 and LC3-II**

Protein lysates were diluted to a final concentration of 0.5 mg/ml and three total protein dilutions were chosen and prepared. For PLIN3 (Fig 1f): 5, 10 and 20 µg; for HSC70 (Fig 1i): 5, 10, 20 µg; for PLIN2 and β-actin (Fig 3a): 2.5, 5 and 10 µg; for LC3-II and GAPDH (Fig 3g): 5, 10, 15 µg. Samples were loaded on 10% 18-well Stain free SDS-gel (#5678034 Bio-Rad Laboratories) or a 8–16% 15-well Stain free SDS-gel (for LC3-II-GAPDH; #4568106 Bio-Rad Laboratories) for electrophoresis, then transferred on low fluorescence PVDF membrane (#1620264 Bio-Rad Laboratories) using the Turbo Transfer system (Bio-Rad Laboratories). Membranes were blocked in 5% BSA in TBST for 1h, then probed for immunoblotting overnight at 4°C. The following primary antibodies (diluted 1:000 in 5% BSA in TBST) were used: PLIN3 (#TA350509 Origene), HSC70 (#NB120–2788 Novus Biologicals), PLIN2 (#TA321278 Origene), LC3 (#NB100–2220 Novus Biologicals), GAPDH (#sc-47724 Santa Cruz). After washing, membranes were incubated for 2h at room temperature with secondaries Goat Anti-Rabbit IgG StarBright Blue 700 (#12004161 Bio-Rad Laboratories) diluted 1:2500 in TBS for PLIN3, HSC70, PLIN2, LC3-II and GAPDH and Anti-Actin hFAB Rhodamine Antibody (#12004163 Bio-Rad Laboratories) diluted 1:5000 in TBS. Signals were captured following antibodies' instructions using a ChemiDoc MP Imaging system with Image Lab Touch Software (Bio-Rad). Signal intensity analysis was performed using the ImageLab software (Bio-Rad Laboratories).

**Immunoblotting of LDLR and LAMP2A**

Hs-HepG2 cell lysates (20, 40, 60 µg protein) were separated on a NuPage 3–8% Tris-Acetate gel (#EA0375PK2 Thermo Fisher Scientific) and then transferred onto nitrocellulose membranes (#88013 Thermo Fisher Scientific). After blocking in StartingBlock™ T20 (TBS) Blocking Buffer (#37543 Thermo Fisher Scientific), the nitrocellulose membranes were incubated overnight at 4°C with polyclonal human LDL-Receptor antibody (1:1000; LS-C146979; LS-Bio). After washing, membranes were incubated for 2h with IRDye® 800CW Goat anti-Rabbit IgG Secondary Antibody (1:40000; 926–32211, LI-COR Biosciences). The specific bands for the LDL-Receptor glycosylated mature form (approx.130 kDa) were detected by Odissey XF imaging system (LI-COR Biosciences). Signal intensity analysis was performed using the ImageJ/Fiji software. The membranes were dried and kept in a plastic container at RT. The membranes were re-hydrated in PBS for 1h at room temperature before being incubated overnight at 4°C with polyclonal human LAMP2A antibody (1:1000; ab18528; Abcam). After washing, membranes were incubated with goat anti-rabbit IgG, (H + L) secondary antibody, HRP diluted 1:10000 in TBST (#32460 Thermo Fisher Scientific). Blots were developed using Pierce ECL Western Blotting Substrate (#10005943 Thermo Fisher Scientific) and a ChemiDoc MP Imaging system with Image Lab Touch Software (Bio-Rad). Signal intensity analysis was performed using the ImageLab software (Bio-Rad Laboratories).

## Supporting information

**S1 Fig. Raw image 1. Original blots for** Fig 1c **and** Fig 2a**.**
(TIF)

**S2 Fig. Raw image 2. Original blots for** Fig 1f **(a) and** Fig 1i **(b).**
(TIF)

**S3 Fig. Normalization to a single dilution point reduces the coefficient of variations given by blot-to-blot variance.** a) t-WB plots of LDLR after normalization of all points for WB1 20 µg point. b) LDLR concentrations from each WB expressed as signal intensity units (AU)/total protein mass (µg) at 20 µg and CV of the concentrations. c) t-WB plots of LDLR after normalization of all points for WB1 40 µg point. d) LDLR concentrations from each WB expressed as signal intensity units (AU)/total protein mass (µg) after normalization at 40 µg and CV of the concentrations. e) t-WB plots of LAMP2A after normalization of all points for WB1 20 µg point. f) LAMP2A concentrations from each WB expressed as signal intensity units (AU)/total protein mass (µg) after normalization at 20 µg and CV of the concentrations. g) t-WB plots of LAMP2A after normalization of all points for WB1 40 µg point. h) LAMP2A concentrations from each WB expressed as signal intensity units (AU)/total protein mass (µg) after normalization at 40 µg. i) Immunoblots of a single lysate of HepG2 cells treated by atorvastatin (5 mM), loaded at 20, 40, 60 µg of total protein onto 4 different WB membranes and probed for the LDLR. j) t-WB plots for LDLR for each WB1–4. k) LDLR concentrations from each WB expressed as signal intensity units (AU)/total protein mass (µg) and CV of the concentrations. l) t-WB plots of LDLR after normalization of all points for WB1 60 µg point. m) LDLR concentrations from each WB expressed as signal intensity units (AU)/total protein mass (µg) after normalization at 60 µg. n) Immunoblots the same 4 WB membranes shown in (i) with LAMP2A antibody. o) t-WB plots for LAMP2A for each WB1–4. p) LAMP2A concentrations from each WB expressed as signal intensity units (AU)/total protein mass (µg) and the overall CV. q) t-WB plots for LAMP2A after normalization of all data points using the WB1–60 µg data point. r) LAMP2A concentrations from each WB expressed as signal intensity units (AU)/total protein mass (µg) and the overall CV.
(TIF)

**S4 Fig. Raw images 3. Original blots for** Fig 2f**.** The blots are shown as composite image (left) and chemiluminescence only (right).
(TIF)

**S5 Fig. Raw images 4. Original blots for** S4 Fig **i.**
(TIF)

**S6 Fig. Raw images 5.** Original blots for S4 Fig n. The blots are shown as composite image (left) and chemiluminescence only (right).
(TIF)

**S7 Fig. Extended immunoblot results from repeated experiments.** a) Immunoblots of PLIN2 and β-actin of THP-1 cells treated with vehicle (DMSO), 25 µg/ml oxLDL, 50 µg/ml oxLDL or 400 µM OA. Each cell lysate sample was loaded at 2.5, 5, 10 µg of total protein per well. Experimental repeats of Fig 3a (n = 3). **b)** Immunoblots of LC3-II and GAPDH of THP-1 cells treated with vehicle (DMSO) or Bafilomycin (+Baf). Each cell lysate sample was loaded at 5, 10, 15 µg of total protein per well. For each WB, a common internal control of THP-1 cells was loaded at 7.5 µg. Experimental repeats of Fig 3g (n = 5).
(TIF)

**S8 Fig. Raw images 6. Original blots for** Fig 3a **(1) and repeats.** The blots are shown as composite image (top left) and fluorescence channels only (bottom left right for PLIN2 and bottom right for β-actin). The representative blots are highlighted in yellow, whereas the repeats are shown in blue.
(TIF)

**S9 Fig. Raw images 7. Original blots for** Fig 3g **(1) and repeats.** The blots are shown as chemiluminescence channels only (top for LC3-II and bottom for GAPDH). The representative blots are highlighted in yellow, whereas the repeats are shown in blue.
(TIF)

**S1 Table. Data analysis for** Fig 1. a) Arbitrary values and t-WB analysis used for schematic in Fig 1b. b) Raw data of LDLR from Fig 1c and t-WB analysis for Fig 1d. c) Arbitrary values and t-WB analysis used for schematic in Fig 1e. d) Raw data of PLIN3 from Fig 1f and t-WB analysis for Fig 1g. e) Arbitrary values and t-WB analysis used for schematic in Fig 1h. f) Raw data of HSC70 from Fig 1i and t-WB analysis for Fig 1j.
(XLSX)

**S2 Table. Data analysis for** Fig 2 **and** Supplementary Fig 1**: LDLR.** a) Raw data of LDLR from Figure 1c and 2a and t-WB analysis for Fig 2b, 2c. b) LDLR signals normalized to WB1 at 20 µg (red) and t-WB analysis for S3 Fig 1a, 1b. c) LDLR signals normalized to WB1 at 40 µg (red) and t-WB analysis for S3 Fig 1c, 1d. d) LDLR signals normalized to WB1 at 60 µg (red) and t-WB analysis for Fig 2d, 2e.
(XLSX)

**S3 Table. Data analysis for** Fig 2 **and** Supplementary Fig 1**: LAMP2A.** a) Raw data of LAMP2A from Figure 2f and t-WB analysis for Fig 2g,2 h. b) LAMP2A signals normalized to WB1 at 20 µg (red) and t-WB analysis for S3 Fig 1e, 1f. c) LAMP2A signals normalized to WB1 at 40 µg (red) and t-WB analysis for S3 Fig 1g, 1h. d) LAMP2A signals normalized to WB1 at 60 µg (red) and t-WB analysis for Fig 2i, 1j.
(XLSX)

**S4 Table. Data analysis for** Supplementary Fig 1**: LDLR and LAMP2A.** a) Raw data of LDLR from S3 Fig 1i and t-WB analysis for S3 Fig 1j, 1k. b) LDLR signals normalized to WB1 at 60 µg (red) and t-WB analysis for S3 Fig 1l, 1m. c) Raw data of LAMP2A from S3 Figure 1n and t-WB analysis for S3 Figure 1o,p. b) LAMP2A signals normalized to WB1 at 60 µg (red) and t-WB analysis for S3 Fig 1q, 1r.
(XLSX)

**S5 Table. Data analysis for** Fig 3**: PLIN2 and β-actin.** a) Raw data of PLIN2 from Figure 3a and t-WB analysis for Fig 3b, 3c. b) Raw data of β-actin from Fig 3a and t-WB analysis for Figure 3d,e. c) PLIN2 normalization to β-actin expressed as fold change compared to control at 2.5 µg protein loading for Fig 3f (left panel). d) PLIN2 normalization to β-actin expressed as fold change compared to control at 5 µg protein loading for Fig 3f (middle panel). e) PLIN2 normalization to β-actin expressed as fold change compared to control at 10 µg protein loading for Fig 3f (right panel).
(XLSX)

**S6 Table. Data analysis for** Fig 3**: LC3-II.** a) Raw data of LC3-II from Fig 3g and t-WB analysis for Fig 3h, 3i. b) LC3-II signals normalized to WB1 internal control 7.5 µg (red) ad t-WB analysis for Fig 3j, 3k.
(XLSX)

**S7 Table. Data analysis for** Fig 3**: GAPDH.** a) Raw data of GAPDH from Fig 3g and t-WB analysis. b) GAPDH signals normalized to WB1 internal control 7.5 µg (red) ad t-WB analysis for Fig 3m, 3n. c) LC3-II normalization to GAPDH expressed at 5 µg protein loading for Fig 3o (left panel). d) LC3-II normalization to GAPDH expressed at 5 µg protein loading for Fig 3o (middle panel). e) LC3-II normalization to GAPDH expressed at 5 µg protein loading for Fig 3o (right panel).
(XLSX)

## Acknowledgments

This work was supported by grants from Karolinska Institutet, Region Stockholm, Swedish Research Council, Swedish Heart and Lung Foundation.

## Data analysis and graphing

Signal intensity analysis was performed with ImageJ/Fiji or ImageLab. Figures and data integration were prepared using GraphPad Prism; t-WB pipeline was created with BioRender.com.

## Author contributions

**Conceptualization:** Matteo Pedrelli, Paolo Parini.

**Data curation:** Alice Maestri.

**Formal analysis:** Alice Maestri, Matteo Pedrelli.

**Funding acquisition:** Paolo Parini.

**Investigation:** Alice Maestri, Olivera Werngren, Maria Olin, Matteo Pedrelli.

**Methodology:** Alice Maestri, Maria Olin, Paolo Parini.

**Project administration:** Ewa Ehrenborg, Paolo Parini.

**Resources:** Ewa Ehrenborg, Paolo Parini.

**Supervision:** Ewa Ehrenborg, Carolina E. Hagberg, Matteo Pedrelli, Paolo Parini.

**Validation:** Olivera Werngren, Maria Olin.

**Visualization:** Alice Maestri.

**Writing – original draft:** Alice Maestri, Matteo Pedrelli.

**Writing – review & editing:** Alice Maestri, Ewa Ehrenborg, Carolina E. Hagberg, Matteo Pedrelli, Paolo Parini.

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
