## [Decision Letter · Decision Letter 0]

27 Mar 2025

Dear Dr. Parini,

Thank you for submitting your manuscript to PLOS ONE. After careful consideration, we feel that it has merit but does not fully meet PLOS ONE’s publication criteria as it currently stands. Therefore, we invite you to submit a revised version of the manuscript that addresses the points raised during the review process.

We look forward to receiving your revised manuscript.

Kind regards,

Yi Cao

Academic Editor

PLOS ONE

Journal Requirements:

2. Thank you for stating the following financial disclosure: [- P.P. 20210622;20200668;20190544 from Swedish Heart-Lung Foundation,

URL:https://www.hjart-lungfonden.se/ - P.P 201602992 from Swedish Research Council, URL: https://www.vr.se/english.html]

4. Please update your submission to use the PLOS LaTeX template. The template and more information on our requirements for LaTeX submissions can be found at http://journals.plos.org/plosone/s/latex.

5. We note that there is identifying data in the Supporting Information file < Suppl Figures, Suppl. Tables, and Extended_Data>. Due to the inclusion of these potentially identifying data, we have removed this file from your file inventory. Prior to sharing human research participant data, authors should consult with an ethics committee to ensure data are shared in accordance with participant consent and all applicable local laws.

-Location data

Please remove or anonymize all personal information (Name), ensure that the data shared are in accordance with participant consent, and re-upload a fully anonymized data set. Please note that spreadsheet columns with personal information must be removed and not hidden as all hidden columns will appear in the published file.

Reviewers' comments:

Reviewer's Responses to Questions

**Comments to the Author**

1. Is the manuscript technically sound, and do the data support the conclusions?

Reviewer #1: Yes

2. Has the statistical analysis been performed appropriately and rigorously?

Reviewer #1: Yes

3. Have the authors made all data underlying the findings in their manuscript fully available?

Reviewer #1: Yes

4. Is the manuscript presented in an intelligible fashion and written in standard English?

Reviewer #1: Yes

Reviewer #1: The authors (Alice et al.) have significantly enhanced the manuscript through extensive revisions, most notably by incorporating additional experimental data that has greatly enriched the study. The corresponding modifications have enhanced the understanding of key points in the article. However, several issues remain that require further attention:

1. While the authors emphasize the error-reduction capability of their method, Figure 2d and 2i show noticeable deviations. Although the first derivative is used for quantification and CV demonstrates reproducibility, the authors should provide quantitative metrics to substantiate their claims of "significant improvements in accuracy and precision," particularly in terms of deviation from true values.

2. Resolution concerns in several results (Fig. 2a and 2f) need clarification. Figure 2a: The absence of WB1 requires explanation. Figure 2a (WB2) and 2f (WB2): The presence of visible bubbles on the far right raises questions about measurement accuracy under such interference. Authors should discuss the reliability of measurements at this lower resolution. Figure 2d and 2i lack legend information. The general trend shows only two points within the linear fitting range, with the third point prone to deviation.

3. Ambiguous descriptions need clarification. E.g: Page 3, line 98: "eliminating many of the errors" - specify which errors (bubbles, background, low resolution, or others)?

4. Statistical concerns: Is using only three points sufficient for slope calculation? Chemical measurements typically require five or more points. Would additional dilution factors and more data points improve precision?

5. Figure optimization: Figures 1 and 3 are overly complex. Consider splitting and simplifying them to highlight core information, limiting to 4-6 subfigures per figure. Less critical information could be moved to supplementary materials. For instance, the two CV comparisons could be consolidated. The current format resembles an experimental report more than a research article.

6. Figure layout: Figure legends should be placed adjacent to their corresponding figures for better readability.

7. Conclusion section: The current text-based claims lack persuasive power. Include at least one concrete case study with quantitative data demonstrating improvement magnitude and deviation from actual results. Present the CV mentioned on page 6 in a more direct and quantifiable manner.

8. Reference formatting: While references have been adjusted, several formatting issues persist:

Page 8, line 280: Citation placement needs correction for proper traceability. Avoid grouping multiple citations together (e.g., page 9, line 306). Inconsistent journal abbreviations: references 6 and 22 use all caps, while others don't. Author name formatting in reference 11 is inconsistent with other entries.

These points should be addressed to strengthen the manuscript's scientific rigor and presentation quality.

**Do you want your identity to be public for this peer review?** For information about this choice, including consent withdrawal, please see our Privacy Policy

Reviewer #1: No

---

## [Author Response · Author response to Decision Letter 1]

22 Apr 2025

Response to Editor and Reviewer comments for manuscript ID: PONE-D-24-33858 “Titration-WB: A methodology for accurate quantitative protein determination overcoming reproducibility errors” by Alice Maestri et al.

Original reviewer comments are shown in blue text, responses in black text. New text added to the revised manuscript based on Reviewer comments is shown with in red.

Editor:

Please ensure that your manuscript meets PLOS ONE's style requirements, including those for file naming. The PLOS ONE style templates can be found at ….

Answer: We thank the editor and we now believe that we have met all the requirements.

2. Thank you for stating the following financial disclosure: [- P.P. 20210622;20200668;20190544 from Swedish Heart-Lung Foundation…

Answer: We have now edited according to what suggested by the editor.

3. PLOS ONE now requires that authors provide the original uncropped and unadjusted images underlying all blot or gel results reported in a submission’s figures or Supporting Information files…

Answer: We have now generated a new file called S1_raw_images with all uncropped Western blots with all the required information. We have also added a reference to the cover letter indicating where these files can be found.

4. Please update your submission to use the PLOS LaTeX template. The template and more information on our requirements for LaTeX submissions can be found at http://journals.plos.org/plosone/s/latex.

Answer: We have now updated our submission to the use of PLOS LaTeX template.

5. We note that there is identifying data in the Supporting Information file < Suppl Figures, Suppl. Tables, and Extended_Data>. Due to the inclusion of these potentially identifying data, we have removed this file from your file inventory…

Answer: We believe that there is a misunderstanding. We do not have any data or indications that potentially identify human being since we have not included any patient in this study. All human material (sera and cell lines) is commercially available and fully anonymized. Hence no need for ethical review by the ethical board.

6. Please review your reference list to ensure that it is complete and correct. If you have cited papers that have been retracted, please include the rationale for doing so in the manuscript text, or remove these references and replace them with relevant current references…

Answer: Based on the comprehensive examination of our refences, we have found no retracted papers in our citation list.

Reviewer #1:

Reviewer #1: The authors (Alice et al.) have significantly enhanced the manuscript through extensive revisions, most notably by incorporating additional experimental data that has greatly enriched the study. The corresponding modifications have enhanced the understanding of key points in the article.

Answer: We thank the reviewer for recognizing the significant improvement of our method on protein measurements and the efforts in making it more understandable and clearer.

However, several issues remain that require further attention:

1. While the authors emphasize the error-reduction capability of their method, Figure 2d and 2i show noticeable deviations. Although the first derivative is used for quantification and CV demonstrates reproducibility, the authors should provide quantitative metrics to substantiate their claims of "significant improvements in accuracy and precision," particularly in terms of deviation from true values.

Answer: We thank the reviewer for this comment. Figure 2 demonstrates the performance of our method in quantifying low-abundance, challenging proteins such as LDLR and LAMP2A, where some variability is expected due to the inherent limitations of the Western blot technique. We have used the coefficient of variation, which reflects both reproducibility and precision—the latter being greater when the coefficient is lower. Thus, a reduction in the coefficient of variation from 36% to 4.2% indicates a significant improvement in precision.

It is difficult to apply the concept of true values to Figure 2. In fact, Figure 2d illustrates that none of the blots are similar to each other in terms of precision (CV = 36%). We could select the data point used for normalization in Figure 2d as a reference value and calculate the error rate and accuracy as the percentage difference from this value. However, we believe that presenting the improvement in accuracy in this manner would unnecessarily complicate the message we wish to convey to the reader—a message that is already clearly supported by the data.

2. Resolution concerns in several results (Fig. 2a and 2f) need clarification.

Figure 2a: The absence of WB1 requires explanation.

Figure 2a (WB2) and 2f (WB2): The presence of visible bubbles on the far right raises questions about measurement accuracy under such interference. Authors should discuss the reliability of measurements at this lower resolution.

Figure 2d and 2i lack legend information. The general trend shows only two points within the linear fitting range, with the third point prone to deviation.

Answer: We are very grateful to the reviewer for raising this point, as it helps to highlight the robustness of our method, particularly its insensitivity to bubble formation—a common issue in Western blotting. The visible bubble in WB2 (Figures 2a and 2f) could indeed compromise quantification if conventional Western blotting were used. However, the t-WB method, which incorporates three dilutions and one-point normalization, enabled us to validate the reliability of the data despite the presence of the bubble. We have added a few sentences to the manuscript to express this concept:

Line196-201: “t-WB also allows us to assess protein quantity in Western blots of lower quality, such as WB2, where a visible bubble is present. While conventional Western blots using a single point would not have allowed us to utilize this data, the use of three total protein dilutions from the same sample and t-WB analysis with one-point normalization demonstrated that the bubble had minimal impact on the quantification of LDLR and LAMP2A (Fig. 2e, j).”

WB1 was not included in Figure 2a as it is already shown in Figure 1c; we have clarified this in the manuscript with the following sentence:

Lines 179-181: “To evaluate the precision of the t-WB method, we compared signals from six independent t-WB blots measuring LDLR expression in HepG2 cell lysates, WB1 shown in Figure 1c and WB2-6 in Figure 2a (Fig. 1c, Fig. 2a-e).”

Finally, we have updated the legends for Figures 2d and 2i to indicate the data used for those graphs.

3. Ambiguous descriptions need clarification. E.g: Page 3, line 98: "eliminating many of the errors" - specify which errors (bubbles, background, low resolution, or others)?

Answer: We have modified the manuscript to reduce ambiguity throughout the text.

4. Statistical concerns: Is using only three points sufficient for slope calculation? Chemical measurements typically require five or more points. Would additional dilution factors and more data points improve precision?

Answer: We thank the reviewer for this valuable comment. While additional dilution points are typically used for analysis of chemical measurements, we have found that using three points offers a practical balance between accuracy and efficiency since only one curve passes through three points. As noted in the discussion, t-WB requires substantial amounts of protein lysate, which may limit the feasibility of multiple experimental runs. Increasing dilution points also significantly raises the number of gels needed. That said, using three points need the certainty that the dilution used are within the linear range of detection as discussed in different points of the manuscript.

5. Figure optimization: Figures 1 and 3 are overly complex. Consider splitting and simplifying them to highlight core information, limiting to 4-6 subfigures per figure. Less critical information could be moved to supplementary materials. For instance, the two CV comparisons could be consolidated. The current format resembles an experimental report more than a research article.

Answer: We appreciate the reviewer’s feedback. As this work introduces a new methodology and as most of the readers will not have the immediate understanding of the matter as the reviewer demonstrates, we believe it is important to present key examples and calculations clearly within the main figures to support reader understanding and a certain degree of redundancy. Figure 1 illustrates three core aspects of t-WB, both theoretically and practically, and we feel it is best kept in its current form. Similarly, Figure 3 highlights a major advantage of our method—its superiority over conventional WB normalization—in two distinct experimental settings. While we understand the concern about figure complexity, we believe presenting these results in a consistent and complete format within the main text better supports comprehension and reproducibility.

6. Figure layout: Figure legends should be placed adjacent to their corresponding figures for better readability.

Answer: We understand this comment and we will make sure with the Editor that figure legends will be adjacent to the corresponding figures at the final formatting stage.

7. Conclusion section: The current text-based claims lack persuasive power. Include at least one concrete case study with quantitative data demonstrating improvement magnitude and deviation from actual results. Present the CV mentioned on page 6 in a more direct and quantifiable manner.

Answer: We sincerely thank the reviewer for this valuable input. We appreciate the suggestion to include more concrete examples and quantitative data to better illustrate the magnitude of improvement and the deviation from actual results. While the results presented in the manuscript are meant to demonstrate the potential of t-WB, our primary aim is to showcase the method’s versatility and robustness across a range of experimental conditions, rather than to provide exhaustive case studies. We recognize that incorporating more detailed data could further strengthen the impact of our findings; however, doing so would shift the focus of the article toward a more experimental report style, potentially detracting from the immediacy and clarity of our main message. In light of the reviewer’s comment, we have revised our conclusion to be more direct and compelling, emphasizing the reduction in inter-experiment variability and the improvement in consistency, even under challenging conditions such as low-quality Western blot runs (e.g., the presence of bubbles or protein smears).

Conclusion

Line 354-379: “The titration-Western Blot (t-WB) method represents a significant advancement in protein quantification, directly addressing and overcoming the persistent limitations of traditional Western blotting. By seamlessly integrating combining titration analysis with regression-based normalization, t-WB establishes a gold standard for accurate, reproducible and truly quantitative protein measurements. Unlike conventional approaches, t-WB eliminates the dependence on housekeeping proteins and neutralizes biases introduced by experimental variability, making it an exceptionally reliable and standardized solution for protein analysis. The use of serial dilutions and regression analysis enables precise quantification, while built-in quality control metrics such as R² values and intercepts (b) help detect and correct experimental inconsistencies. These features provide a level of accuracy and control that distinguishes t-WB from conventional methods.

Our validation across diverse experimental conditions has shown that t-WB not only reduces inter-experiment variability, but also significantly enhances cross-blot comparisons and uncovers subtle biological changes that traditional normalization methods routinely miss. Remarkably, t-WB maintains its high performance even with low-quality Western blot runs, such as those compromised by bubbles or protein smears, underscoring its robustness and adaptability in real-world laboratory settings.

By decisively overcoming the shortcomings of classical Western blot protocols, t-WB redefines the standards for quantitative protein analysis. It equips researchers with a powerful, trustworthy tool that guarantees superior accuracy and reproducibility, fundamentally advancing the reliability and impact of molecular biology research. As scientific methodologies continue to evolve, t-WB stands as a foundational innovation, bridging the gap between qualitative detection and precise quantitative analysis. Its widespread adoption will not only elevate the rigor and reproducibility of scientific studies but also drive deeper, more reliable insights into the intricate dynamics of protein expression and regulation. The t-WB method is poised to become an indispensable asset in the pursuit of scientific excellence.”

8. Reference formatting: While references have been adjusted, several formatting issues persist:

Page 8, line 280: Citation placement needs correction for proper traceability. Avoid grouping multiple citations together (e.g., page 9, line 306). Inconsistent journal abbreviations: references 6 and 22 use all caps, while others don't. Author name formatting in reference 11 is inconsistent with other entries.

Answer: We apologize for these formatting oversights. We have corrected all citations both in-text and in the references section. We would like to highlight that references 6 and 22 use all caps as the journal name is “ELECTROPHORESIS” and that the author’s name formatting in reference 11 is formatted in the same way as the others. For all references we have used EndNote with style Vancouver and imported our entries as downloaded from PUBMED as .ris files.

---

## [Decision Letter · Decision Letter 1]

6 May 2025

Titration-WB: A methodology for accurate quantitative protein determination overcoming reproducibility errors

PONE-D-25-04655R1

Dear Dr. Parini,

We’re pleased to inform you that your manuscript has been judged scientifically suitable for publication and will be formally accepted for publication once it meets all outstanding technical requirements.

Kind regards,

Yi Cao

Academic Editor

PLOS ONE

Additional Editor Comments (optional):

Reviewers' comments:

Reviewer's Responses to Questions

**Comments to the Author**

Reviewer #1: (No Response)

2. Is the manuscript technically sound, and do the data support the conclusions?

Reviewer #1: Yes

3. Has the statistical analysis been performed appropriately and rigorously?

Reviewer #1: Yes

4. Have the authors made all data underlying the findings in their manuscript fully available?

Reviewer #1: Yes

5. Is the manuscript presented in an intelligible fashion and written in standard English?

Reviewer #1: Yes

Reviewer #1: The authors have addressed the major concerns raised previously; however, the figures have not been revised, which significantly limits the clarity and accessibility of the manuscript, especially for readers outside the field. While the core content may meet the journal’s basic requirements, the lack of improvement in visual presentation remains a notable weakness. If the editor thinks the current version acceptable, I will support the acceptance of this manuscript.

**Do you want your identity to be public for this peer review?** For information about this choice, including consent withdrawal, please see our Privacy Policy

Reviewer #1: No

---

## [Editor Report · Acceptance letter]

PONE-D-25-04655R1

PLOS ONE

Dear Dr. Parini,

I'm pleased to inform you that your manuscript has been deemed suitable for publication in PLOS ONE. Congratulations! Your manuscript is now being handed over to our production team.

Kind regards,

on behalf of

Dr. Yi Cao

Academic Editor

PLOS ONE